# FROM POINTS TO FUNCTIONS: INFINITE-DIMENSIONAL REPRESENTATIONS IN DIFFUSION MODELS

**Sarthak Mittal**[†,1,2] **, Guillaume Lajoie**[1,2]**, Stefan Bauer**[5]**, Arash Mehrjou**[3,4]
[1]Mila, [2]Université de Montréal, [3]MPI-IS, [4]ETH Zurich, [5]KTH Stockholm

## ABSTRACT

Diffusion-based generative models learn to iteratively transfer unstructured noise to a complex target distribution as opposed to Generative Adversarial Networks (GANs) or the decoder of Variational Autoencoders (VAEs) which produce samples from the target distribution in a single step. Thus, in diffusion models every sample is naturally connected to a random trajectory which is a solution to a learned stochastic differential equation (SDE). Generative models are only concerned with the final state of this trajectory that delivers samples from the desired distribution. Abstreiter et al. (2021) showed that these stochastic trajectories can be seen as continuous filters that wash out information along the way. Consequently, there is an intermediate time step at which the preserved information is optimal for a given downstream task. In this work, we show that a combination of information content from different time steps gives a strictly better representation for the downstream task. We introduce an attention and recurrence based modules that "learn to mix" information content of various time-steps such that the resultant representation leads to superior performance in downstream tasks.

## 1 INTRODUCTION

Diffusion-based models (Sohl-Dickstein et al., 2015; Song et al., 2020; 2021; Sajjadi et al., 2018; Niu et al., 2020; Cai et al., 2020; Chen et al., 2020a; Saremi et al., 2018; Dhariwal & Nichol, 2021; Luhman & Luhman, 2021; Ho et al., 2021; Mehrjou et al., 2017) are generative models that apply step-wise perturbations to the samples of the data distribution (eg. CIFAR10), modeled via a Stochastic Differential Equation (SDE), until convergence to some prior unstructured distribution (eg. $\mathcal{N}(\mathbf{0}, \mathbf{I})$). In contrast to this diffusion process, a score model is learned to approximate the reverse process that iteratively converges to the data distribution from the prior distribution mentioned before. In this work, we follow Abstreiter et al. (2021) which augments such diffusion-based systems with an encoder for learning a representation that can be used for downstream tasks.

## 2 BEYOND FIXED REPRESENTATIONS

We first outline how diffusion-based representation learning systems are trained. Given some example $\mathbf{x}_0 \in \mathbb{R}^d$ which is sampled from the target distribution $p_0$, the diffusion process constructs the trajectory $(\mathbf{x}_t)_{t \in [0,1]}$ through the application of an SDE. In this work, we consider the Variance Exploding SDE (Song et al., 2021) for this diffusion process, defined as

$$d\mathbf{x} = f(\mathbf{x}, t) + g(t)d\mathbf{w} := \sqrt{\frac{d[\sigma^2(t)]}{dt}} d\mathbf{w} \tag{1}$$

where $\mathbf{w}$ is the standard Wiener process and $\sigma^2(\cdot)$ the noise variance of the diffusion process. This leads to a closed form distribution of $\mathbf{x}_t$ conditional on $\mathbf{x}_0$ as $p_{0t}(\mathbf{x}_t|\mathbf{x}_0) = \mathcal{N}(\mathbf{x}_t; \mathbf{x}_0, [\sigma^2(t) - \sigma^2(0)]\mathbf{I})$. Given this diffusion process modeled through the Variance Exploding SDE, the reverse SDE takes a similar form but requires the knowledge about the score function, i.e. $\nabla_{\mathbf{x}} \log p_t(\mathbf{x})$ for

---

[†]Correspondence author sarthmit@gmail.com

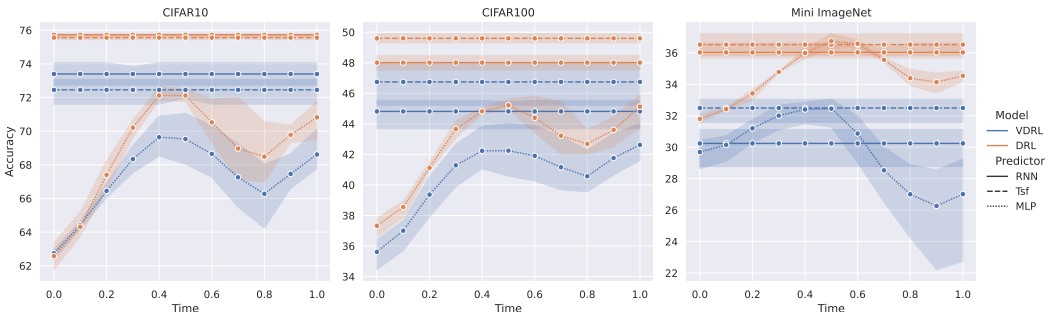

Figure 1: Downstream performance of single point based representations (MLP) and full trajectory based representations (RNN and Tsf) on different datasets for both types of learned encoders: probabilistic (VDRL) and deterministic (DRL).

all $t \in [0, 1]$. A common way to obtain this score function is through the Explicit Score Matching (Hyvärinen & Dayan, 2005) objective,

$$\mathbb{E}_{\mathbf{x}_t} \left[ \left\| s_\theta(\mathbf{x}_t, t) - \nabla_{\mathbf{x}_t} \log p_t(\mathbf{x}_t) \right\|^2 \right] \tag{2}$$

which suffers from just one hiccup, which is that data about the ground-truth score function is not available. To solve this problem, Denoising Score Matching (Vincent, 2011) was proposed,

$$\mathbb{E}_{\mathbf{x}_0} \left[ \mathbb{E}_{\mathbf{x}_t | \mathbf{x}_0} \left[ \left\| s_\theta(\mathbf{x}_t, t) - \nabla_{\mathbf{x}_t} \log p_{0t}(\mathbf{x}_t | \mathbf{x}_0) \right\|^2 \right] \right] \tag{3}$$

where the term $\log p_{0t}(\mathbf{x}_t | \mathbf{x}_0)$ is available due to its closed-form structure. Given that the above objective cannot be reduced to 0, Abstreiter et al. (2021) proposes the objective

$$\mathbb{E}_{\mathbf{x}_0} \left[ \mathbb{E}_{\mathbf{x}_t | \mathbf{x}_0} \left[ \left\| s_\theta(\mathbf{x}_t, E_\phi(\mathbf{x}_0, t), t) - \nabla_{\mathbf{x}_t} \log p_{0t}(\mathbf{x}_t | \mathbf{x}_0) \right\|^2 \right] \right] \tag{4}$$

where the additional input $E_\phi(\mathbf{x}_0, t)$ to the score function is obtained from a learned encoder and provides information about the unperturbed sample that might prove useful for denoising data at time step $t$ in the diffusion trajectory. Training of this system can lead to the objective being reduced to 0, thereby providing incentive to the encoder $E_\phi(\cdot, t)$ to learn meaningful representations for each time $t$. From this, we obtain a trajectory-based representation $(E_\phi(\mathbf{x}_0, t))_{t \in [0,1]}$ from each sample $\mathbf{x}_0$ as opposed to just a finite sized representation obtained from typical Autoencoder (Bengio et al., 2013; Vinyals et al., 2016; Kingma & Welling, 2013; Rezende et al., 2014) and Contrastive Learning (Chen et al., 2020b; Grill et al., 2020; Caron et al., 2021; Bromley et al., 1993; Chen & He, 2020) based approaches.

## 2.1 INFINITE-DIMENSIONAL REPRESENTATION OF FINITE-DIMENSIONAL DATA

Normally in autoencoders or other *static* representation learning methods, the input data $\mathbf{x}_0 \in \mathbb{R}^d$ is mapped to a single point $\mathbf{z} \in \mathbb{R}^c$ in the code space. However, our proposed algorithm learns a richer representation where the input $\mathbf{x}_0$ is mapped to a curve in $\mathbb{R}^c$ instead of a single point through the encoder $E_\phi(\cdot, t)$. Hence, the learned code is produced by the map $\mathbf{x}_0 \to (E_\phi(\mathbf{x}_0, t))_{t \in [0,1]}$ where the infinite-dimensional object $(E_\phi(\mathbf{x}_0, t))_{t \in [0,1]}$ is the encoding for $\mathbf{x}_0$.

The learned code is at least as good as static codes in terms of separation induced among the codes. Consider two input samples $\mathbf{x}_0$ and $\mathbf{x}_0'$, hence we have:

$$\left\| E_\phi(\mathbf{x}_0, 0) - E_\phi(\mathbf{x}_0', 0) \right\| \leq \sup_{t \in [0,1]} \left\| E_\phi(\mathbf{x}_0, t) - E_\phi(\mathbf{x}_0', t) \right\| \tag{5}$$

which implies that the downstream task can at least recover the separation provided by finite-dimensional codes from the infinite-dimensional code by looking for the maximum separation along the representation trajectory.

A downstream task can leverage this rich encoding in various ways. Consider the classification task where we want to find a mapping $f : \mathbb{R}^d \to \{0, 1\}$ from input data to the label space. Instead of giving $\mathbf{x}_0$ as the input to $f$, we define $f : \mathcal{H} \to \{0, 1\}$ where the input to the classifier is the whole trajectory $(E_\phi(\mathbf{x}_0, t))_{t \in [0,1]}$. Thus, the classifier can now use RNN and Transformer models to make use of the information content of the entire trajectories.

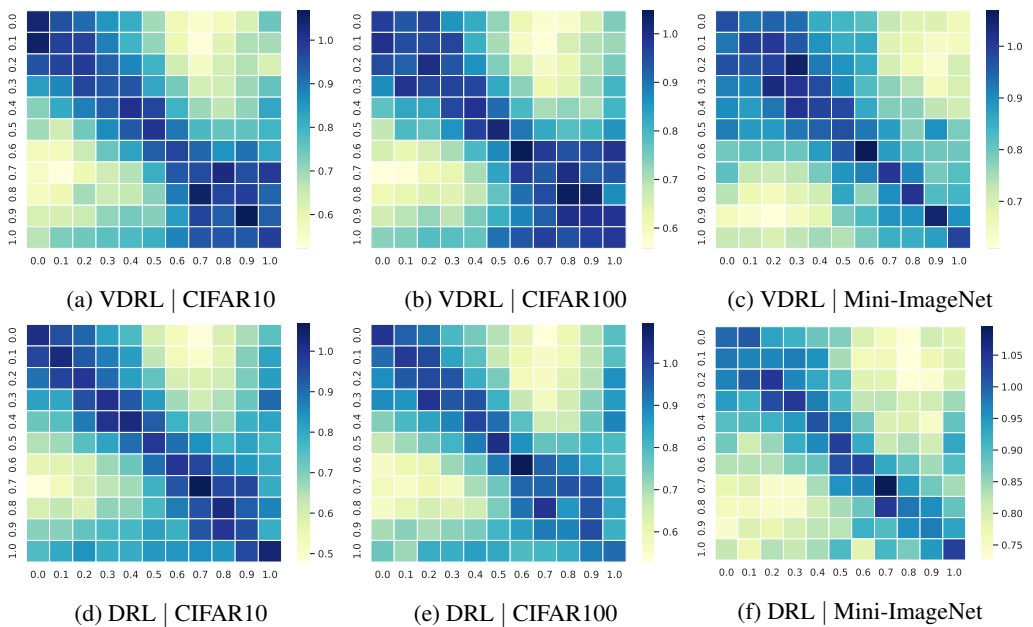

Figure 2: Normalized Mutual Information between different points on the trajectory. Cell $(i, j)$ demonstrates the normalized mutual information, estimated with the MINE algorithm, between the representations at time $t = i$ and $t = j$.

## 3 EXPERIMENTS

We first train two kinds of diffusion-based generative model as outlined in Abstreiter et al. (2021), based on probabilistic (VDRL) and deterministic (DRL) encoders respectively. After training, the encoder model is kept fixed. For all our downstream experiments, we use this trained encoder to obtain the trajectory based representation for each of the samples. While the trajectories lie in a continuous domain $[0, 1]$, we sample it at a regular intervals with length $0.1$. This leads to a discretization of the trajectory, which is then used for various analysis as outlined below.

### 3.1 DOWNSTREAM PERFORMANCE REVEALS BENEFITS OF TRAJECTORY INFORMATION

To understand the benefits of utilizing the trajectory-based representations, we train standard Multi-Layer Perceptron (MLP) models at different points on the trajectory and compare it with Recurrent Neural Network (RNN) (Hochreiter & Schmidhuber, 1997; Cho et al., 2014) and Transformer (Vaswani et al., 2017) based models that are able to aggregate information from different parts of the trajectory.

We evaluate the MLP, RNN and Transformer based downstream models on diffusion systems with both probabilistic encoders (VDRL) and also non-probabilistic ones (DRL). In Figure 1, we see the performance of these different setups for the following datasets: CIFAR10 (Krizhevsky et al., a), CIFAR100 (Krizhevsky et al., b) and Mini-ImageNet (Vinyals et al., 2016). We typically see that RNN and Transformer based models perform better than even the peaks obtained by the MLP systems. This shows that there is no single point on the trajectory that encapsulates all the information stored in the trajectory, and thus utilizing the whole trajectory as opposed to individual points leads to an improvement in performance.

### 3.2 MUTUAL INFORMATION REVEALS DIFFERENCES ALONG THE TRAJECTORY

In an effort to understand whether different parts of the trajectory based representation actually contain different types of information about the sample, we evaluate the mutual information between the representations at various points in the trajectory. We use the MINE algorithm (Belghazi et al., 2018) to estimate the mutual information between the representations at any two different points in the trajectory. Through this algorithm, we compute and analyse a normalized version of the

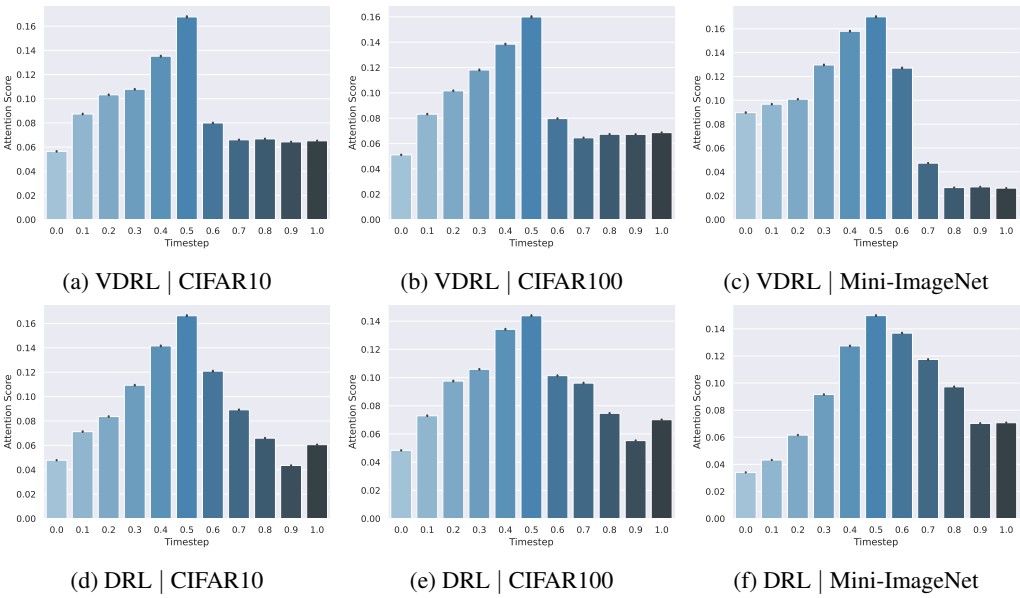

Figure 3: Attention Scores provided to different points on the trajectories, which are obtained from diffusion based representation learning systems with probabilistic encoders (VDRL; top row) and with deterministic encoders (DRL; bottom row) across the following datasets (a) Left: CIFAR10, (b) Middle: CIFAR100, and (c) Right: Mini-Imagenet.

mutual information, defined as $\text{NMI}(\mathbf{X}; \mathbf{Y}) := \text{I}(\mathbf{X}; \mathbf{Y})/\sqrt{\text{H}(\mathbf{X})\text{H}(\mathbf{Y})}$ where $\text{I}(\cdot \, ; \, \cdot)$ is the standard Mutual Information function (Cover, 1999) and $\text{H}(\cdot)$ is the entropy function.

Figure 2 illustrates the normalized mutual information between representations at different parts of the trajectory across three different datasets: CIFAR10, CIFAR100 and Mini-ImageNet as well as two different types of models: VDRL and DRL, where the former uses a probabilistic encoder and the latter doesn't. We see large normalized mutual information values near the principal diagonal and small values that are away from it, demonstrating that nearby representations on the trajectory are similar whereas distant points in the trajectory are considerably different. This shows that different parts of the trajectory learn to encode different kinds of information.

### 3.3 ATTENTION REVEALS RELEVANCE OF DIFFERENT PARTS OF THE TRAJECTORY

To complement the analysis in Sections 3.1 and 3.2, we train a single-layered Transformer model for downstream prediction, which comes from a learned embedding that queries information from different parts of the trajectory. Through the analysis of the attention scores at different points in the trajectory, we realize that the middle parts of the trajectory are the most important, as illustrated in the high attention scores around $t = 0.5$ in Figure 3.

This is in line with the performance results in Figure 1 which also shows that amongst the single-point MLP-based systems, the best downstream performance is reached near the middle of the trajectory.

### 4 CONCLUSION

Through our analysis, we realize that the encoder $E_\phi(\cdot, t)$ actually learns different kinds of information at different time-steps $t$. Typically the mid-points of the trajectory are the most important for downstream classification tasks but we uncover that using the whole trajectory, which is a discretization of an infinite-dimensional object, is much better than just singular points on it. What kind of semantic information is encoded in the different parts of the trajectories? Can we leverage the full infinite-dimensional object without heuristic based discretizations? Our aim is to not only analyze the nature of information encoded in different parts of these trajectories but also leverage them for downstream tasks without heuristic discretizations.

ACKNOWLEDGEMENTS

SM would like to acknowledge the support of UNIQUE and IVADO towards his research. GL acknowledges the support from Canada CIFAR AI Chair Program, Samsung SAIT, and NSERC Discovery Grant [RGPIN-2018-04821].

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
