# OpenReview forum: "From Points to Functions: Infinite-dimensional Representations in Diffusion Models"
_ICLR.cc/2022/Workshop/DGM4HSD — ICLR 2022 DGM4HSD workshop Poster_

### Official Review · Reviewer_o5Xr · 2022-03-19
**Good contribution to diffusion-based models studies, but improvement is needed**

**Rating:** 7
**Confidence:** 4

**Review:**

The paper proposes to leverage intermediate representations appearing in the diffusion models for obtaining more informative latent representations. Authors experimentally demonstrate that intermediate representations from different time steps contain different information and that downstream tasks can benefit from using these representations.

The paper is written well, the goals and contributions are correctly stated, and all the claims are supported by empirical evidence and some theoretical intuition. The study highlights importance of considering intermediate representations of diffusion-based models, which is a good contribution to this rapidly developing field.

Comments:
1. The overall accuracy of classification tasks is low compared to common medium-size architectures. For example, Wide ResNet achieves about 96% on CIFAR-10 ([https://paperswithcode.com/sota/image-classification-on-cifar-10?tag_filter=3](https://paperswithcode.com/sota/image-classification-on-cifar-10?tag_filter=3)) while in the paper the best model gives about 76%.  This raises a question whether the observed effect holds for non-toy modes, which are expressive enough to solve the task well in a traditional setting, i.e. without using intermediate representations.
2. Following the previous point, it is not clear from the paper how models’ sizes correspond to each other and whether the superiority is actually given by leveraging intermediate states, not by a larger capacity. More details about architectures are needed to make sense of comparison fairness and for reproducibility.
3. According to equation for NMI in section 3.2, the corresponding matrix should be symmetric, i.e. NMI(x, y) = NMI(y, x). However, this doesn’t hold for matrices in Fig.2, e.g. cell(1.0, 0.6) =/= cell(0.6, 1.0) in subfigure (f). It would be worth to explain this discrepancy in the text or check whether a mistake happened in NMI computation/plotting
4. It would be good to provide some intuition why the middle parts of the trajectory are the most informative, as it is shown in the Figure 3.
5. In section 2.1. authors provide a good motivation why it is worth to consider the whole trajectory instead of a single point, but it is not clear how the encoder helps. Why do authors decide to use encoded trajectory E_phi(x_0, t) instead of original x_t?

Recommended decision: Accept, but I would highly recommend considering the comments above to improve the paper.

---

### Official Review · Reviewer_Aye2 · 2022-03-23
**From Points to Functions: Infinite-Dimensional Representations in Diffusion Models**

**Rating:** 5
**Confidence:** 3

**Review:**

### Summary
The paper studies representation learning using score based generative models. It uses the representation learning objective from the earlier work Abstreiter et al where they introduce a trainable encoder  $E_\phi$ in the denoising score-based function training. This encoder yields an infinite dimensional representation for the input $x_0$. The authors then discretize the time steps in uniformly spaced intervals to obtain a sequence representation for a single datapoint instead of a single latent vector of fixed dimension. The main takeaways from the paper are

- Firstly, RNN and Transformer based models are better at using this sequence based representation than MLP based models trained at each point.
- Secondly, nearby values on the trajectory capture similar information whereas distant points are different.
- Lastly, based on attention values obtained from training a single layered Transformer network they see that the middle parts of the trajectory being most important.

These contributions should be better called out towards the beginning of the paper.

### Overview

The paper analyzes the quality of representations learned through a score-based formulation from the prior work of Abstreiter et al.  This is studied in the context of an image classification problem on both probabilistic encoder VDRL and non-probabilistic one (DRL) on CIFAR10, CIFAR100 and Mini-Image-Net. The paper doesn't describe how $E_\phi$ is parameterized, is it a neural network - if yes , what is the architecture ?

 For CIFAR10 and CIFAR100 both transformer and RNN based models perform better than MLP based models trained at different points along the trajectory but for mini-image net which is a harder task RNN doesn't perform that well and the transformer is at best similar in performance to the best performance along the trajectory for MLP based model. This is not addressed in the paper could benefit with some discussion around this. The technical contribution of the paper seems limited. The findings from the three experiments are somewhat anticipated.

My overall recommendation at this point is a “reject”. Though, I think the paper can benefit from further deep-diving.
   Overall, the paper is short and easy to follow , however more details around related work could be included.


### Misc (not part of review)

- Curious to hear your thoughts on how something like this could be extended to text
- The reverse U plot for performance of single MLP based classifiers trained at different points in the trajectory is very interesting. What could be the underlying reason behind the points in the middle of the trajectory being more important for the class prediction?

---

### Decision · Program_Chairs · 2022-03-28

Accept (Poster)